# The fluoride permeation pathway and anion recognition in Fluc family fluoride channels

Benjamin C McIlwain[1], Roja Gundepudi[2], B Ben Koff[1], Randy B Stockbridge[1,2]*

[1]Department of Molecular, Cellular, and Developmental Biology, University of Michigan, Ann Arbor, United States; [2]Program in Biophysics, University of Michigan, Ann Arbor, United States

**Abstract** Fluc family fluoride channels protect microbes against ambient environmental fluoride by undermining the cytoplasmic accumulation of this toxic halide. These proteins are structurally idiosyncratic, and thus the permeation pathway and mechanism have no analogy in other known ion channels. Although fluoride-binding sites were identified in previous structural studies, it was not evident how these ions access aqueous solution, and the molecular determinants of anion recognition and selectivity have not been elucidated. Using x-ray crystallography, planar bilayer electrophysiology, and liposome-based assays, we identified additional binding sites along the permeation pathway. We used this information to develop an oriented system for planar lipid bilayer electrophysiology and observed anion block at one of these sites, revealing insights into the mechanism of anion recognition. We propose a permeation mechanism involving alternating occupancy of anion-binding sites that are fully assembled only as the substrate approaches.

## Introduction

Microbes are protected from the cytoplasmic accumulation of environmental fluoride ion (F[-]) by export of the toxic anion via fluoride channels known as Flucs (*Baker et al., 2012*; *Ji et al., 2014*; *McIlwain et al., 2021*). These small, homodimeric ion channels are remarkable proteins in two regards: first, their unusual 'dual-topology' architecture, in which the two subunits of the homodimer are arranged antiparallel with respect to each other (*Stockbridge et al., 2014*; *Stockbridge et al., 2013*), yielding a double-barreled pair of pores related by twofold symmetry (*Stockbridge et al., 2015*; *Last et al., 2016*; *Turman et al., 2015*; *Turman and Stockbridge, 2017*); second, the Flucs stand out among anion channels for their extreme substrate selectivity (*Stockbridge et al., 2013*). In contrast to most characterized families of anion channels, which tend to be non-selective among anions and sometimes poorly discriminate against cations, the Flucs are arguably the most selective ion channels known, with >10,000-fold selectivity against the biologically abundant chloride (*Stockbridge et al., 2013*). This extreme selectivity prevents collapse of the membrane potential due to chloride or cation leak through the Fluc channels, which are constitutively open. Among anion channels, the stringent selectivity displayed by the Flucs is atypical. Most characterized anion channels handle the most abundant ion in their milieu, usually chloride ion (Cl[-]), and other halides and pseudohalides that might compete with the physiological ion are present at much lower concentrations.

Crystal structures of representative Fluc channels from *Bordetella pertussis* (Fluc-Bpe) and an *Escherichia coli* virulence plasmid (Fluc-Ec2) provide an opportunity to understand the molecular basis for anion permeation in the Flucs (*Stockbridge et al., 2015*; *McIlwain et al., 2018*). The protein possesses two deep, aqueous vestibules with an electropositive character due to an absolutely conserved arginine sidechain and a deeply buried sodium ion at the center of the

*For correspondence:
stockbr@umich.edu

protein (*Turman and Stockbridge, 2017*; *McIlwain et al., 2020*). The structures captured four elec-tron densities assigned as fluoride ions, two in each pore, positioned near the center of the protein, at some distance from the vestibules. These ions are aligned along the polar face of TM4, referred to as the polar track. They are located 6–10 Å from the aqueous solution, with no clear aqueous pathway leading to the external solution. Mutation of the sidechains that coordinate the proposed fluoride ions inhibits fluoride throughput but does not alter the ion selectivity of these proteins (*Stockbridge et al., 2015*; *Last et al., 2017*). Thus, characterizing the rest of the fluoride permeation route is the first step toward identifying the residues responsible for fluoride ion recognition.

Here we combine x-ray crystallography, planar lipid bilayer electrophysiology, and liposome flux assays to identify access points to the polar track, including a non-specific anion-binding site at the bottom of the aqueous vestibule. We propose that fluoride ions accumulate in this electropositive vestibule before entering the fluoride-selective region of the pore, reprising a familiar feature of many ion channels. After traversing the polar track, the fluoride ions then emerge at another point in the opposite vestibule on the opposite side of the membrane, near a conserved glutamate that plays a role in discriminating against Cl$^-$.

## Results

### Anions enter the fluoride pathway through the electropositive vestibule

The electropositive vestibule, lined with conserved, polar sidechains, is an obvious candidate for fluoride entry into the channel. Spherical, non-protein electron densities were observed in this region, but without additional evidence of anionic character, they were assigned as water molecules (*Stockbridge et al., 2015*). To test whether any of these densities might better be assigned as anions, we endeavored to crystallize Fluc channels with bromide (Br$^-$), an anomalous scatterer. We were unable to generate diffracting Fluc-Bpe crystals in the presence of Br$^-$, but we were successful in solving the structure of Fluc-Ec2 in the presence of 100 mM Br$^-$ (*Table 1*).

Anomalous difference maps show two prominent peaks, located in equivalent, non-crystallo-graphic symmetry-related positions at the bottom of the aqueous vestibules (*Figure 1A*). These den-sities are coordinated by a sidechain that is invariant among Fluc channels, Ser81, along with the highly conserved Thr82 (*Figure 1B*, upper panel). In maps from previous Fluc-Bpe structures (*Stockbridge et al., 2015*), a positive density occupies this same position between the homologous hydroxyl sidechains (*Figure 1B*, lower panel). This site is exposed to bulk water in the vestibule but is likely to be partially dehydrated, with aliphatic sidechains, including Ile48 in close proximity to the bound bromide ion (*Figure 1C*).

In order to test whether this anion-binding site is part of the fluoride permeation pathway, we introduced an I48C mutation to Fluc-Ec2 and assessed the effect of modification by a bulky, anionic, thiol-reactive reagent, (2-sulfonatoethyl)methanethiosulfonate (MTSES), on fluoride conduction. We performed these experiments on a C74A background, which behaves like the wild-type (WT) protein in F$^-$ efflux assays (*Figure 1—figure supplement 1*). A second cysteine in the native Ec2 sequence, C16, cannot be altered without destabilizing the protein. However, this residue is buried at the inter-face of helices 1 and 1′, and does not react with thiol reagents in the folded protein. In planar lipid bilayers, I48C mediates robust fluoride currents, which rapidly diminish by ~50% upon addition of saturating MTSES to the cis chamber (*Figure 1D*), consistent with full modification of a cis-facing thiol in a population of channels with oppositely oriented pores. In contrast, MTSES addition to channels with WT I48 (C74A background) does not alter the fluoride currents (*Figure 1D*). We sought to recreate the MTSES block experiment in Fluc-Bpe channels, but we did not observe effi-cient labeling of a cysteine introduced at the corresponding position, Ile50. However, mutation of Ile50 in Bpe to a bulkier tryptophan sidechain diminished the rate of fluoride transport by ~400-fold in liposome efflux experiments, possibly by sterically hindering fluoride access to the bottom of the vestibule (*Figure 1—figure supplement 2*).

In order to probe the role of the anion-coordinating sidechains in more detail, we mutated Fluc-Ec2's bromide-coordinating Ser81 to alanine, threonine, or cysteine, and also constructed a S81A/T82A double mutant. For all four mutants, we measured fluoride channel activity using either single-channel electrophysiology or bulk liposome efflux experiments (*Figure 2A and B*), and we solved

**Table 1.** Crystallography data collection and refinement statistics.

| | Ec2-WT | Ec2-S81A | Ec2-S81C | Ec2-S81A/T82A | Ec2-S81T |
|---|---|---|---|---|---|
| Data collection | | | | | |
| Space group | $P4_1$ | $P4_1$ | $P4_1$ | $P4_1$ | $P4_1$ |
| Cell dimensions | | | | | |
| a, b, c (Å) | 87.6, 87.6, 144 | 87.4, 87.4, 141.9 | 87.2, 87.2, 142.7 | 87.5, 87.5, 147.4 | 87.1, 87.1, 145.2 |
| α, β, γ (°) | 90, 90, 90 | 90, 90, 90 | 90, 90, 90 | 90, 90, 90 | 90, 90, 90 |
| Resolution (Å) | 34.4–3.11 (3.3–3.11) | 39.1–2.5 (2.6–2.5) | 46.7–2.9 (3.0–2.9) | 41.9–3.1 (3.3–3.1) | 28.4–2.7 (2.8–2.7) |
| $R_{merge}$ | 0.491 (2.31) | 0.140 (1.846) | 0.363 (3.437) | 0.723 (6.147) | 0.217 (2.104) |
| $R_{pim}$ | 0.203 (0.938) | 0.057 (0.742) | 0.156 (1.434) | 0.290 (2.446) | 0.088 (0.833) |
| Mn $I/\sigma I$ | 7.2 (2.0) | 11.9 (1.7) | 9.8 (2.5) | 8.5 (2.1) | 10.3 (2.0) |
| $CC_{1/2}$ | 0.996 (0.61) | 0.998 (0.61) | 0.98 (0.59) | 0.998 (0.73) | 0.998 (0.71) |
| Completeness (%) | 99.85 (100) | 99.5 (100) | 99.83 (100) | 99.85 (99.95) | 99.8 (100) |
| Multiplicity | 13.3 (13.9) | 13.7 (14.1) | 12.5 (13.0) | 13.6 (14.0) | 13.8 (14.4) |
| Refinement | | | | | |
| Resolution | 33.3–3.11 | 37.68–2.5 | 46.65–2.9 | 39.11–3.1 | 28.28–2.7 |
| No. of reflections | 19,500 | 36,591 | 23,580 | 20,055 | 29,192 |
| $R_{work}$/$R_{free}$ | 23.7/27.6 | 24.0/25.2 | 22.3/25.8 | 23.0/25.2 | 21.9/25.6 |
| Ramachandran favored | 93.3 | 96.5 | 95.9 | 94.7 | 96.1 |
| Ramachandran outliers | 0.23 | 0.46 | 0.23 | 0.23 | 0.46 |
| r.m.s. deviations | | | | | |
| Bond length (Å) | 0.005 | 0.002 | 0.002 | 0.008 | 0.008 |
| Bond angle (°) | 0.653 | 0.532 | 0.489 | 0.782 | 0.934 |
| PDB code | 7KKR | 7KKA | 7KKB | 7KK9 | 7KK8 |

Statistics for the highest-resolution shell are shown in parentheses. r.m.s., root-mean-square.

x-ray crystal structures of the mutants together with Br⁻ (*Figure 2C*). No other halides or pseudohalides were present in crystallization solutions.

The functional experiments showed that fluoride throughput is inhibited in these mutants. S81A had a mild effect, with a ~50% decrease in conductance to 3.7 ± 0.4 pS (seven single-channel measurements), compared to 7 pS for the WT protein (*Stockbridge et al., 2013*; *Figure 2B*). The S81A/T82A double mutant had a more severe effect, with fluoride throughput diminished to 8530 ± 30 s⁻¹, a ~100-fold reduction in the rate (*Figure 2A*). In accord with the fluoride transport experiments, a strong anomalous peak persisted in the S81A structure but was weaker in the S81A/T82A double mutant (*Figure 2C*). In both cases, the densities shifted away from the channel center toward the external solution, moving about 2 Å closer to the vestibule Arg22s.

The S81C and S81T phenotypes were more extreme: for both mutants, fluoride efflux from liposomes was completely abolished (*Figure 2A*). From the structural data, it is not readily apparent why S81T does not transport fluoride. A bromide density is observed in a similar position, coordinated by the threonine's hydroxyl, and with similar intensity as wild-type, and the surrounding residues are not perturbed by this mutation.

In contrast, the structure of S81C provides a possible explanation for the lack of fluoride transport observed in the liposome flux assays. An anomalous density is present in the vestibule, but has moved ~2 Å farther up into the aqueous vestibule, relative to the Br⁻ position in the WT protein (*Figure 2D*). We posit that the electropositive environment of the vestibule perturbs the cysteine pKₐ such that it is deprotonated at the pH of these experiments (pH 9 in the crystal structure and pH 7.5 in the liposome flux experiments). The pKₐ prediction software PropKa reinforces this possibility, calculating an approximate pKₐ value of 6 for S81C in the crystal structure of this mutant (*Bas et al., 2008*). To test this idea explicitly, we monitored currents mediated by Ec2 S81C in planar lipid bilayers as a function of changing pH. Whereas fluoride currents were near zero at pH 7.4, currents

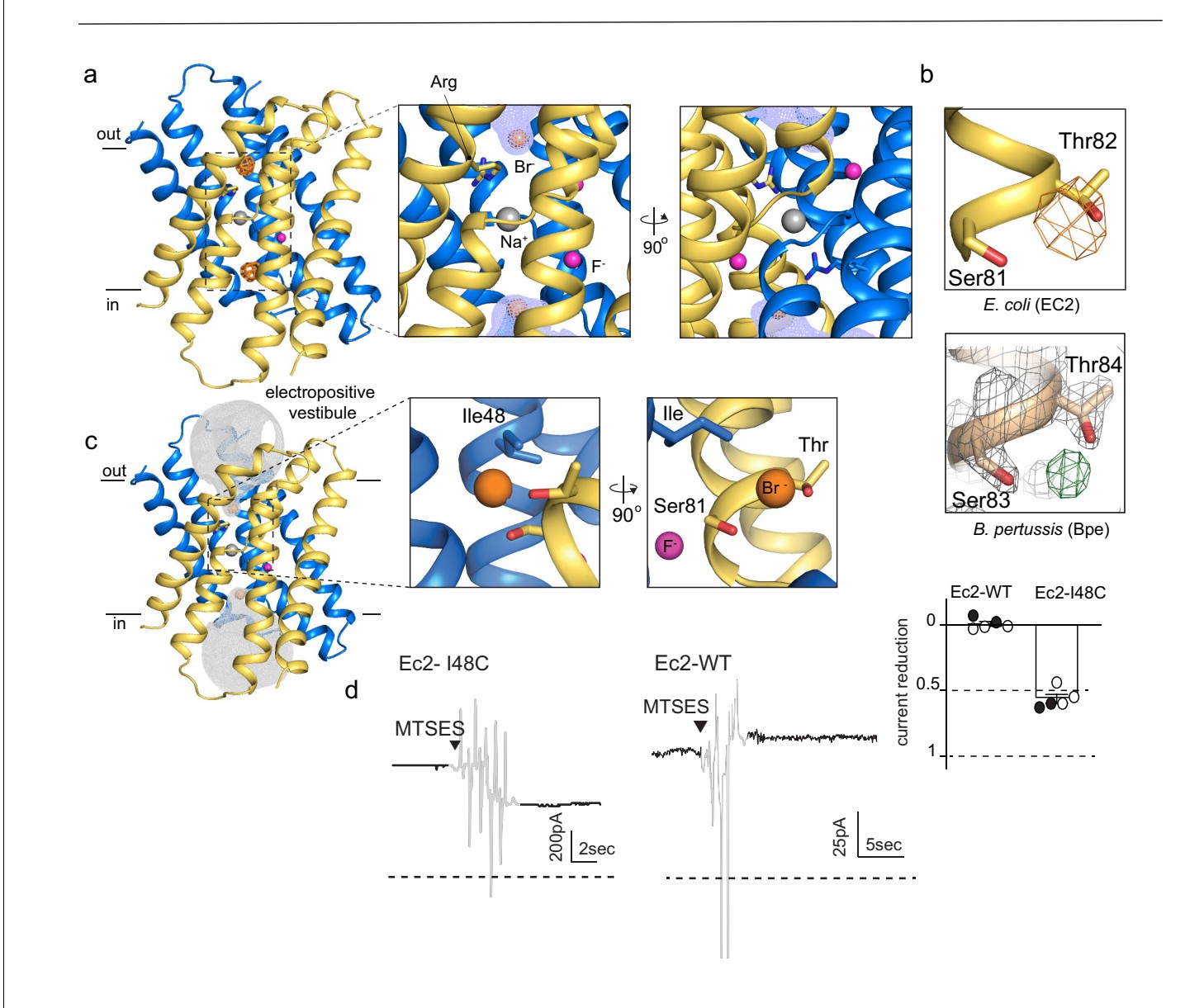

**Figure 1.** An anion-binding site in the Fluc channel vestibule. (**a**) Structure of Fluc-Ec2 with Br⁻. Monomers are shown in maize and blue, with fluoride ions as pink spheres, sodium as a gray sphere, and anomalous difference map shown as an orange mesh, contoured at 5σ. Zoomed-in views depict Br⁻ as orange spheres, with the aqueous vestibule indicated by a blue mesh and vestibule arginines shown as sticks. (**b**) Comparison of vestibule anion-binding site for Fluc-Ec2 (top) and Fluc-Bpe (bottom; PDB: 5NKQ). For Fluc-Ec2, the Br⁻ anomalous difference map is displayed as an orange mesh and contoured at 5σ. For Fluc-Bpe, the $F_o$-$F_c$ map is displayed in green and contoured at 3σ. $2F_o$-$F_c$ electron density is shown for sidechains and displayed as a gray mesh, contoured at 2σ. (**c**) Additional views of the Br⁻-binding site in Fluc-Ec2, with Ile48, Ser81, and Thr82 shown as sticks. (**d**) Electrical recordings for multichannel bilayers of Fluc-Ec2 I48C and wild-type (WT) Fluc-Ec2. Dashed line indicates the zero-current level. Saturating (2-sulfonatoethyl)methanethiosulfonate (MTSES) was added at the indicated time. Regions of the recording with electrical noise from mixing are colored light gray to assist with figure interpretation. Traces are representative of data collected from five independent bilayers. Right panel, normalized current after MTSES addition. Replicates from two independent preps are shown in black or white. Average current change for Ec2 I48C upon MTSES addition (mean ± SEM from five bilayers): 56 ± 3% decrease. Current change for Ec2 WT upon MTSES addition (mean ± SEM from five bilayers): 0.7 ± 1.7% increase.

The online version of this article includes the following source data and figure supplement(s) for figure 1:

**Source data 1.** Measurements of current decrease upon MTSES addition.

**Source data 2.** Fluoride efflux measurements for Bpe-I50W.

**Figure supplement 1.** Fluoride efflux from Fluc-Ec2 C74A (blue and green traces) or WT Fluc-Ec2 channel (black trace) proteoliposomes.

*Figure 1 continued on next page*

*Figure 1 continued*

**Figure supplement 2.** Fluoride efflux from Fluc-Bpe I50W (blue trace) or WT Fluc-Bpe channel (black trace) proteoliposomes.

increased dramatically when the pH was decreased to 5.5 (*Figure 2E*, *Figure 2F*, *Figure 2—figure supplement 1*). The increase in F⁻ currents was fully reversible with pH, and WT activity was not altered by changing pH over this range. The analogous mutation in Fluc-Bpe channels, S83C, exhibits similar pH sensitivity (*Figure 2E*, *Figure 2F*, *Figure 2—figure supplement 1*).

Taken together, these experiments show that the anion-binding site at the bottom of the vestibule is on the fluoride permeation pathway . This anion-binding site is located immediately adjacent to one of the fluoride ions in the polar track, and we imagine fluoride ions enter the vestibule, become dehydrated, before eventually being stripped of water entirely as the ion is translocated from the bottom of the vestibule to the polar track. Translocation between the vestibule and the polar track must contribute to anion selectivity since the bromide anomalous density is observed in the former location, but never in the latter. However, we could not detect any change in chloride transport by these mutants (*Figure 2—figure supplement 2*), motivating us to search for additional pore-lining sidechains on the opposite end of the pore.

## A trio of sidechains defines the opposite end of the pore

To identify additional pore-lining sequences, we began by analyzing the sequences of the eukaryotic relatives of the homodimeric bacterial Flucs, known as Fluoride Export proteins (*Li et al., 2013*). Whereas the bacterial Flucs assemble as dual-topology homodimers with a pair of symmetry-related pores, the eukaryotic fluoride channels are expressed as a two-domain single polypeptide with a linker helix that enforces antiparallel topology of the domains (*Smith et al., 2015*). In the FEX proteins, this ancient fusion event has permitted drift of redundant sequences, including degradation of one of the two pores (*Berbasova et al., 2017*). A clear pattern has been identified in which residues that line one pore (mostly, but not entirely, from the C-terminal domain) are highly conserved, whereas the corresponding residues from the second, vestigial pore (mostly, but not entirely, from the N-terminal domain) have drifted (*Stockbridge et al., 2015*; *Smith et al., 2015*; *Berbasova et al., 2017*). We reasoned that the other amino acids that follow this pattern of conservation and degradation might be expected to also contribute to the pore.

We selected representative eukaryotic FEX proteins from yeasts and plants, and aligned the N- and C-terminal domains with the sequence of Fluc-Bpe in order to identify residues that follow eukaryotic pore conservation patterns (*Figure 3A*). We chose Fluc-Bpe for this analysis rather than Fluc-Ec2, because Fluc-Bpe has higher sequence homology to the eukaryotic FEX domains. We identified three additional residues that follow the same pattern of conservation as other pore-lining sequences: a threonine, a tyrosine, and a glutamate (blue highlighting in alignment). In the Fluc-Bpe structure, the three homologous sidechains (Thr39, Glu88, and Tyr104) associate within hydrogen-bonding distance of each other, one contributed by each pore-lining helix, TM2, TM3, and TM4. They are positioned near the protein's aqueous vestibules, and Tyr104 is also within hydrogen-bonding distance of a fluoride ion within the pore (*Figure 3B*).

These residues are well-conserved among Flucs more generally (*Macdonald and Stockbridge, 2017*). From an alignment of all homodimeric Fluc sequences in the PFAM database (*Mistry et al., 2021*), we found that Thr39 is conserved in ~95% of the sequences we studied, Glu88 is conserved in >85% of sequences (~10% Asp and ~ 5% Gln), and Tyr104 is conserved in 55% of sequences (~35% Asn and ~15% Ser). The strong conservation of these residues across multiple kingdoms, the asymmetric distribution among eukaryotic domains that is consistent with other pore-lining sequences, and their close spatial relationship with one another motivated further functional analysis of this molecular triad.

T39 and Y104 proved sensitive to mutagenesis, and only conservative mutations were permitted at these positions. Using bulk liposome efflux assays as a binary measurement of F⁻ transport, we found that we obtained transport-competent mutants when Thr39 was replaced by Ser, but not Val, Asn, Ala, or Cys (*Figure 3—figure supplement 1*). When Tyr104 was replaced by Phe, robust fluoride efflux activity was observed, but mutants with Ser, His, or Ile in this position all had anemic fluxes in the range of 100 ions/s (*Figure 3—figure supplement 1*). Glu88 was somewhat more

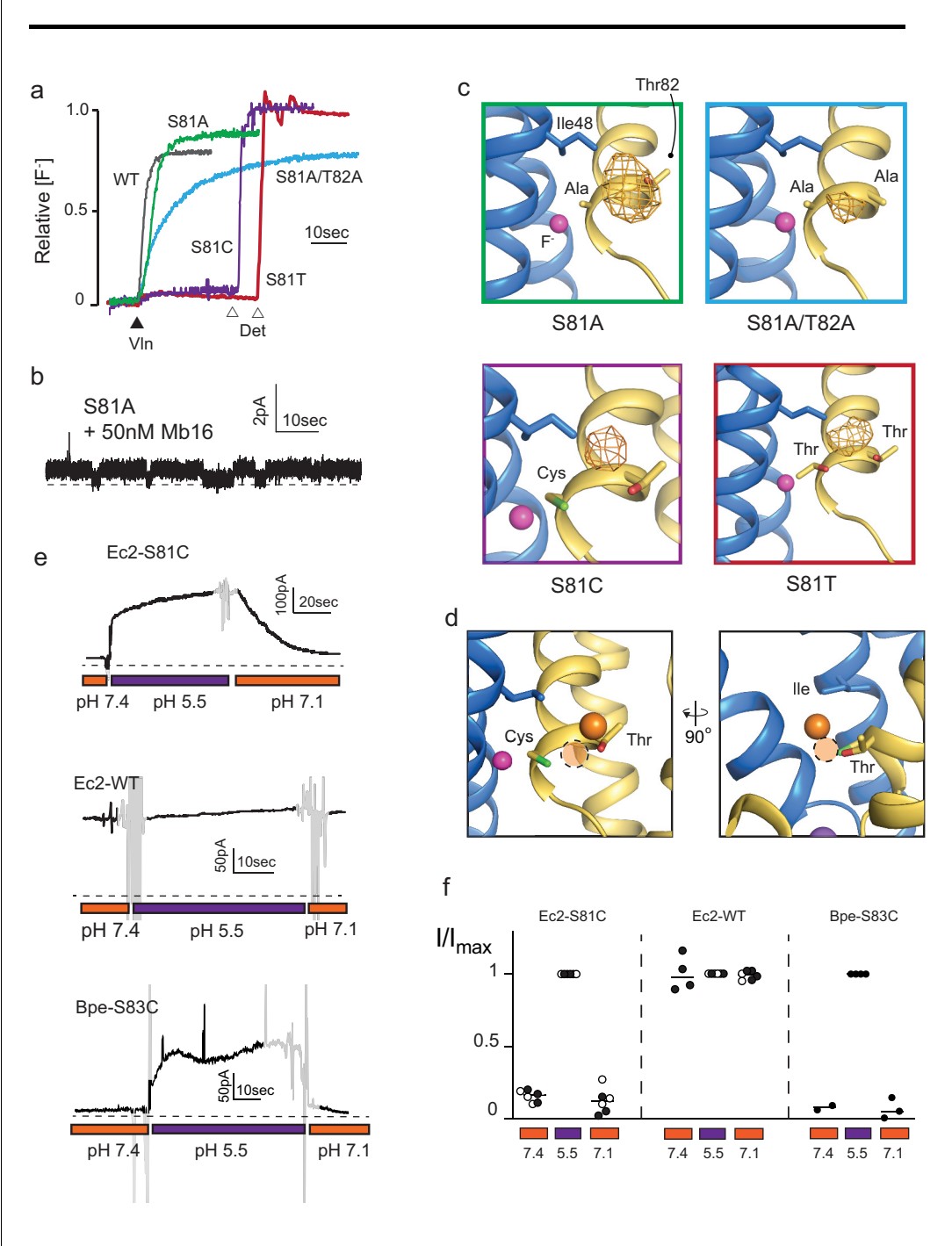

**Figure 2.** Mutagenesis of vestibule anion-binding site. (**a**) Fluoride efflux from liposomes monitored with a fluoride-selective electrode: wild-type (WT) Fluc-Ec2 (gray), S81A (green), S81A/T82A (blue), S81T (red), and S81C (purple). Efflux initiated by the addition of valinomycin (black triangle). After reaching steady state, the remaining encapsulated fluoride was released by detergent addition (open triangles). Each trace is normalized against total encapsulated fluoride. Traces are representative of results from at least two independent biochemical purifications. Results from all replicates are tabulated in *Table 5*. (**b**) Representative single-channel recording of S81A in the presence of a blocking monobody to identify the zero-current level (dashed line). (**c**) Bromine anomalous difference maps for S81A, S81A/T82A, S81T, and S81C contoured at 5σ. The frame around each panel is colored as in panel (**a**). (**d**) Comparison of the position of Br⁻ density in S81C (orange sphere) and WT Ec2 (dashed orange circle). (**e**) Fluoride currents mediated by Ec2-S81C, WT Ec2, and Bpe-S83C channels. pH was adjusted during the experiment as indicated. Regions of the recording with electrical noise from mixing are colored light gray to assist with figure interpretation. Traces are representative of recordings from three to six independent bilayers. Additional replicate traces can be found in *Figure 2—figure supplement 1*. (**f**) Summary of all replicates of experiments shown in panel (**e**). Values are

*Figure 2 continued on next page*

*Figure 2 continued*

normalized against the maximum steady-state current (5 s average) measured at pH 5.5 for that trace. Black and white points indicate different protein preparations.

The online version of this article includes the following source data and figure supplement(s) for figure 2:

**Source data 1.** Fluoride efflux data.
**Source data 2.** Single-channel measurements of S81A.
**Source data 3.** Normalized currents for Ec2-S81C, Ec2, and Bpe-S83C as a function of pH.
**Source data 4.** Electrophysiological recordings of Ec2-S81C, Ec2, and Bpe-S83C as a function of pH.
**Source data 5.** Chloride efflux data.
**Figure supplement 1.** Additional replicate traces for the experiment in *Figure 2E*.
**Figure supplement 2.** Chloride efflux experiments with fluoride-transporting Fluc-Ec2 S81A/T82A and the homologous Fluc-Bpe S83A/T84A (blue traces).

permissive: Ala, Asp, and Gln were all tolerated, but not Lys (*Figure 3—figure supplement 1*). To experimentally probe whether Glu88 is in the anionic carboxylate form or protonated at pH 7, we performed bilayer experiments in which we recorded currents at pH 7 and then raised the pH in a stepwise fashion to 8.7. We observed reduced fluoride currents as pH was increased, but the difference in these effects between channels bearing Glu and Gln at position 88 was minimal (*Figure 3—figure supplement 2*). Since changing the protonation state of an acidic sidechain along the permeation pathway would be expected to have substantial ramifications for fluoride currents, these experiments suggest that the protonation state of Glu88 does not change as the pH is increased from 7 to 8.7 and, therefore, that the $pK_a$ of E88 falls below ~6.5 or above ~9. A $pK_a$ perturbation of a glutamate to >9 would be quite unusual, and we therefore argue that it is more likely that Glu88 is not protonated at physiological pH. In agreement with this interpretation, Propka calculates an approximate $pK_a$ for Glu88 of 5.7 (*Bas et al., 2008*).

Those triad mutants that permitted fluoride transport in efflux assays were also assessed using single-channel electrophysiology (*Figure 3C,D*). T39S, E88D, and Y104F retained $F^-$ conductance to at least 75% of WT levels, and we do not interpret these differences as mechanistically important. In contrast, E88Q exhibited currents one-fifth of the WT levels, a more substantial difference that is also statistically significant at $p < 0.0001$ (unpaired t-test). T39S and Y104F both showed differences in dynamic behavior compared to WT Fluc-Bpe proteins, which are constitutively open and show no closures or sub-conductance states. T39S undergoes long periods of robust throughput ($\tau_o = 9.2 \pm 0.2$ s), punctuated by brief channel closures ($\tau_c = 35.3 \pm 0.4$ ms) (*Figure 3D*). Y104F was more dynamic, with shorter open intervals ($\tau_o = 1.9 \pm 0.2$ s and $\tau_c = 33.2 \pm 2.5$ ms) (*Figure 3D*, inset). Thus, single-channel recordings suggest that one role of the triad is to stabilize the three pore-lining helices in an open, fluoride-conducting conformation. Upon addition of channel-binding monobodies (*Stockbridge et al., 2014*; *Turman and Stockbridge, 2017*), familiar current block is observed, indicating that despite the increased conformational flexibility, the structure of the channel is not perturbed to a significant extent (*Figure 3—figure supplement 3*).

## Anion recognition at the triad

None of the fluoride-conductive mutants constructed thus far transport chloride ion, as probed using our most sensitive metric of chloride transport, liposome efflux assays (*Figure 4—figure supplement 1*). However, we noticed that halides and pseudohalides inhibit fluoride currents with a wide range of potencies (*Figure 4A*, *Table 2*). The recognition series ($OCN^- > SCN^- > NO_3^- > Cl^-$) deviates from common determinants of anion selectivity, such as anion radius, $\Delta G_{hydration}$, $\Delta G_{Born}$, or the lyotropic (Hofmeister) series (*Table 3*, *Figure 4—figure supplement 2*). In these titrations, full inhibition of the fluoride currents is not achieved. The inhibitory effects are best fit by a two-site-binding isotherm, with weak binding to a second site (*Table 2*). Because the Fluc channel possesses a pair of antiparallel pores, the observed behavior might reflect anion interactions at both the vestibule and triad sides of the channel. In order to separate the effects of anion block at these two positions, and to better quantify the affinity, we exploited the S83C vestibule mutant described in *Figure 2E* by recording channels under asymmetric pH conditions. The cis side of the bilayer was maintained at pH 7.5, silencing any pore with a cis-facing vestibular S83C. The trans side of the bilayer was

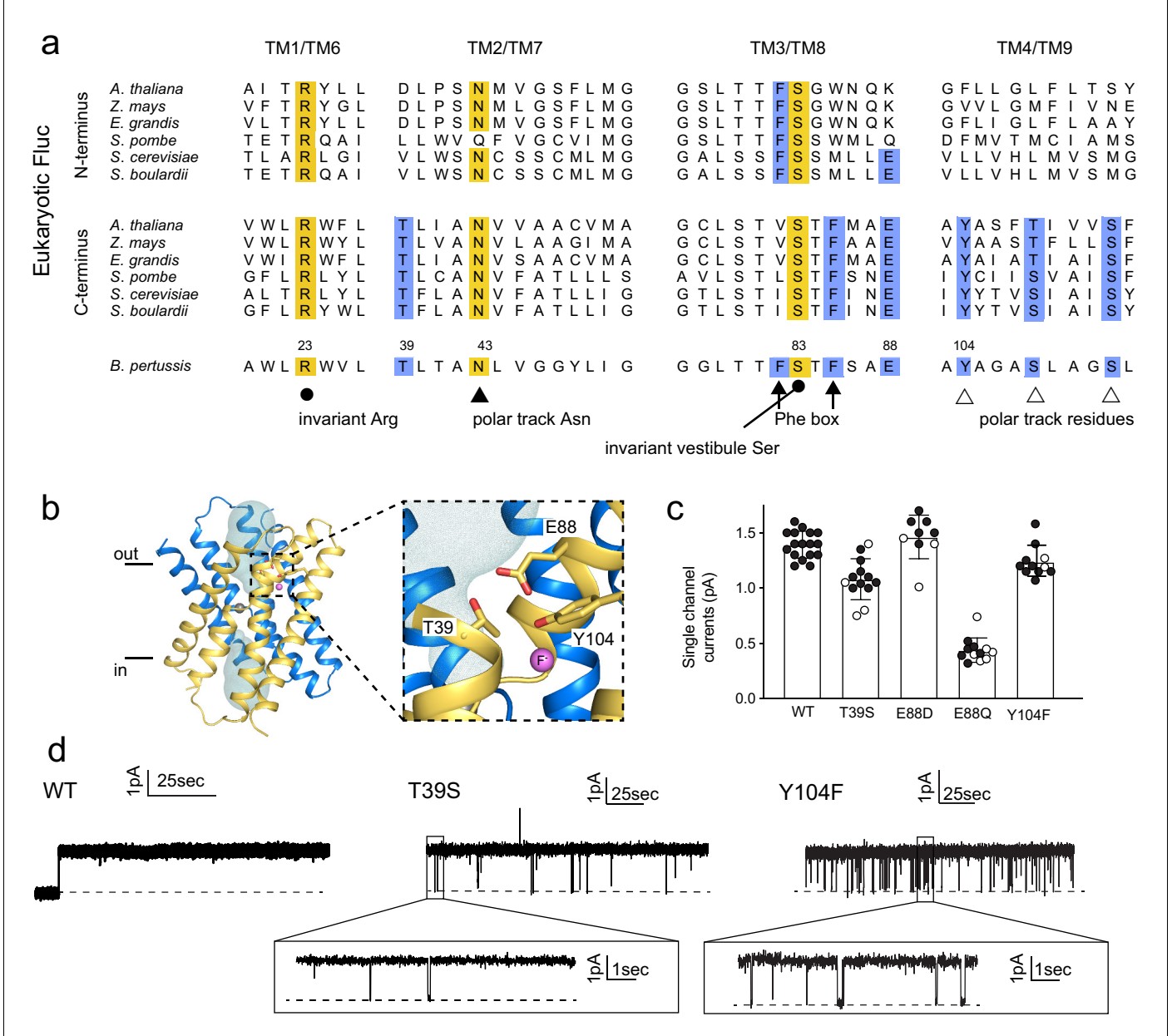

**Figure 3.** Identification and characterization of triad residues. (**a**) Sequence alignment of Fluc-Bpe with N- and C-terminal domains of representative eukaryotic fluoride channels (transmembrane helices only). Invariant pore-lining residues are shown in yellow. Pore-lining residues that are conserved in only one pore of the eukaryotic FEX channels are highlighted in blue. Residue numbering from Fluc-Bpe is shown (note that S83 in Fluc-Bpe is equivalent to S81 in Fluc-Ec2). (**b**) Structure of Fluc-Bpe (PDB: 5NKQ) with triad residues indicated as sticks, aqueous vestibule as a mesh, and fluoride ions as pink spheres. (**c**) Single-channel currents for wild-type (WT) Fluc-Bpe and indicated mutants measured at a holding voltage of 200 mV. Error bars represent the mean and SEM. Black and white points indicate different protein preparations. (**d**) Representative single-channel electrophysiological recordings for WT Fluc-Bpe, Bpe T39S, and Bpe-Y104F.

The online version of this article includes the following source data and figure supplement(s) for figure 3:

**Source data 1.** Single-channel current measurements.

**Source data 2.** Single-channel recordings of Bpe-WT, Bpe-T39S, and Bpe-Y104F.

**Source data 3.** Fluoride efflux data.

**Source data 4.** pH dependence of WT and E88Q (oriented channels).

**Source data 5.** Monobody block of T39S and Y104F.

**Figure supplement 1.** Fluoride efflux traces with indicated Fluc-Bpe mutants.

**Figure supplement 2.** pH dependence of Fluc-Bpe E88 and E88Q.

**Figure supplement 3.** Monobody block of currents mediated by Fluc-Bpe mutants T39S and Y104F.

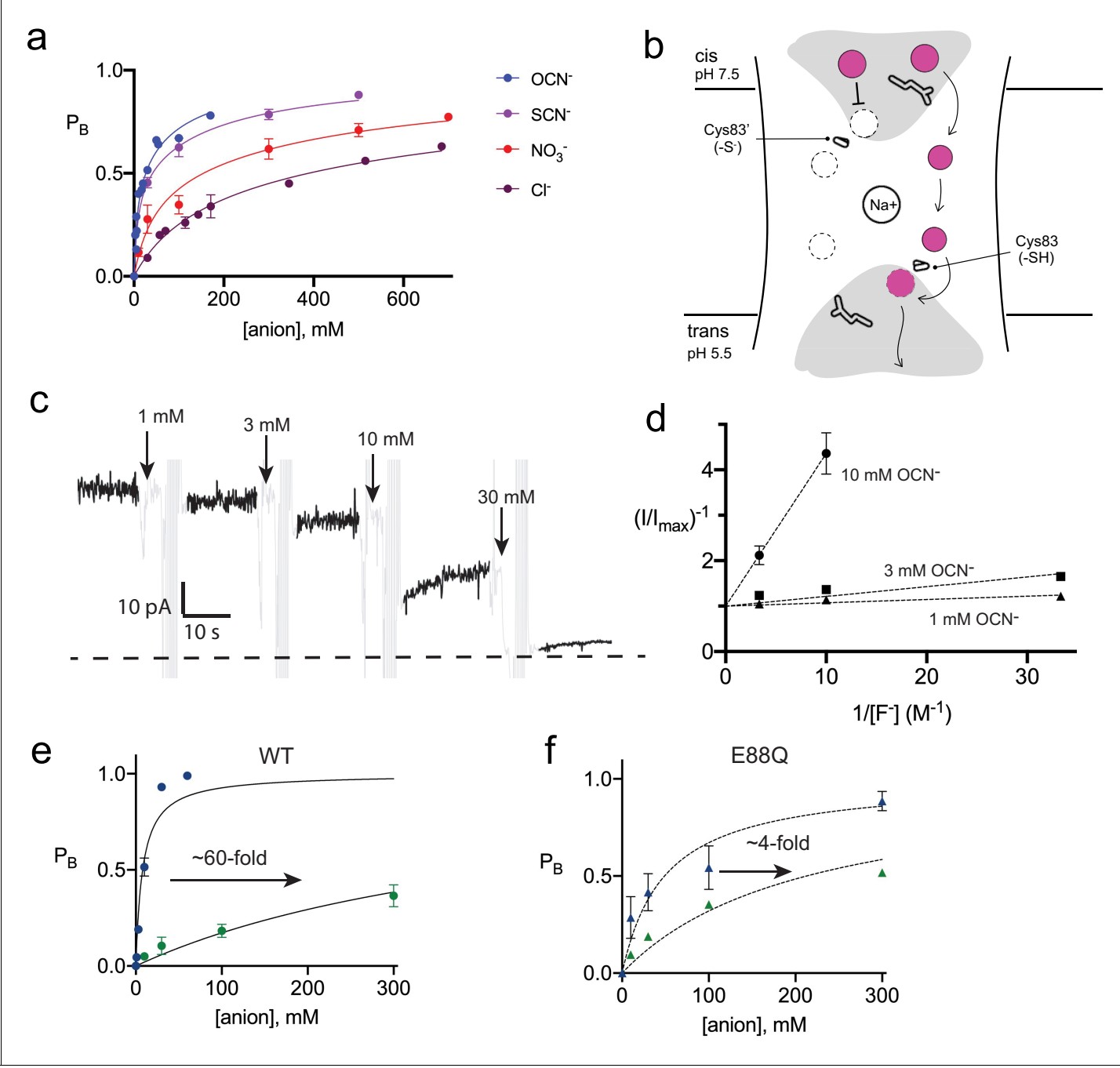

**Figure 4.** Inhibition of Fluc-Bpe and Fluc-Bpe E88Q currents by halides and pseudohalides. (**a**) Fraction of blocked current as a function of anion addition. The solid lines represent fits to a two-site-binding isotherm, constrained so that the maximum $P_B$ for each site is 0.5. In this model, anions bind to single sites that are located on opposite sides of the dual-topology pores. $K_i$ values for fits are reported in *Table 2*. Data are collected from three independent bilayers. Where present, error bars represent SEM of independent replicates. (**b**) Cartoon of strategy for orienting Bpe channels for anion block experiments. Gray area indicates aqueous vestibules. Sidechains E88 and S83C are shown as sticks. (**c**) Representative electrical recording showing $OCN^-$ addition to fluoride currents mediated by oriented Bpe-S83C channels. The zero-current level is indicated with a dashed line. Cyanate additions are indicated by the arrows. Regions of the recording with electrical noise from cyanate addition and mixing are colored light gray to assist with figure interpretation. (**d**) Lineweaver-Burke analysis of $OCN^-$ block as a function of $F^-$ concentration. Dashed lines represent linear fits to the data. All measurements were performed in triplicate from independent bilayers; where not visible, error bars are smaller than the diameter of the point. (**e, f**) Fraction of blocked current in S83C (**e**) or S83C/E88Q (**f**) oriented channels as a function of anion addition. Points and error bars represent the mean and SEM from three independent bilayers. Where not visible, error bars are smaller than the diameter of the point. Solid lines represent fits to a single-

*Figure 4 continued on next page*

*Figure 4 continued*

site-binding isotherm with $P_{B,max}$ = 1. $K_i$ values from fits are reported in *Table 2*. Comparison of replicate measurements from independent preps are shown in *Figure 4—figure supplement 3*.

The online version of this article includes the following source data and figure supplement(s) for figure 4:

**Source data 1.** Current block by anion addition (dual-topology channels).
**Source data 2.** Current block by anion addition (oriented channels).
**Source data 3.** Current blocked by OCN⁻ addition as a function of F⁻ concentration.
**Source data 4.** Chloride efflux traces for triad mutants.
**Source data 5.** Light-scattering traces.
**Source data 6.** SCN⁻ block of dual-topology channels.
**Figure supplement 1.** Chloride efflux experiments for fluoride-transporting Fluc-Bpe mutants Y104F, T39S, E88Q, and E88A.
**Figure supplement 2.** Halide and pseudohalide block of Fluc-Bpe.
**Figure supplement 3.** Prep-to-prep comparison of Cl⁻ and OCN⁻ titrations to bilayers with E88Q and WT channels (S83C oriented system).
**Figure supplement 4.** Light-scattering experiments to detect OCN⁻ permeation through Fluc-Bpe channels.
**Figure supplement 5.** Representative fluoride current recordings with OCN⁻ and Cl⁻ titrations.
**Figure supplement 6.** Titration of SCN⁻ into WT (a) and E88Q (b) bilayers.

adjusted to pH 5.5, so that pores with a trans-facing vestibular S83C retained WT-like function (*Figure 4B*).

With this oriented system, we tested the effect of OCN⁻ and Cl⁻ addition to the cis (pH 7.5) side of the bilayer, isolating anion interactions at the side of the pore defined by the T-E-Y triad. In OCN⁻ titration experiments, currents were reduced almost to the zero-current level at 30 mM OCN⁻, showing that the higher affinity OCN⁻-binding site is on the triad side of the pore (*Figure 4C*). Using the oriented system, we performed OCN⁻ addition experiments in the presence of 30–300 mM F⁻. The apparent affinity of OCN⁻ increased as F⁻ concentration decreased, showing that binding and inhibition at the triad site is competitive with fluoride (*Figure 4D*). For both OCN⁻ and Cl⁻, block of the fluoride currents was well approximated by a single-site-binding isotherm that saturates at full inhibition, although we did not perform experiments at the ~molar Cl⁻ concentrations that would be required to fully block currents (*Figure 4E*, *Figure 4—figure supplement 3*, *Table 2*). In contrast, fits to the data with the two-site-binding model used for the dual-topology WT channels were poor. Under our usual experimental conditions with 300 mM F⁻, a fit to a single-site-binding isotherm yielded $K_i$ values of ~400 mM for chloride, the most abundant biological halide, and 8 mM for OCN⁻, in very good agreement with the value estimated from the dual-topology WT channels (*Table 2*). Although OCN⁻ blocked Fluc-Bpe with relatively high affinity, liposome flux experiments showed that OCN⁻ is not permeant (*Figure 4—figure supplement 4*).

It is notable that one of the participants in the triad, E88, is itself an anion. In order to understand the interplay between the E88 carboxylate and the blocking anions, we mutated E88 to glutamine

**Table 2.** Fit parameters for anion block experiments.

| | Dual-topology channels | | | | Oriented channels | | | |
|---|---|---|---|---|---|---|---|---|
| | WT/OCN⁻ | WT/SCN⁻ | WT/NO₃ | WT/Cl⁻ | WT/OCN⁻ | WT/Cl⁻ | E88Q/OCN⁻ | E88Q/Cl⁻ |
| $K_{i,1}$ | 6.8 mM | 9.0 mM | 45 mM | 137 mM | 7.9 mM | 480 mM | 48.9 mM | 213 mM |
| $B_{max1}$ | 0.5 | 0.5 | 0.5 | 0.5 | 1.0 | 1.0 | 1.0 | 1.0 |
| $K_{i,2}$ | 100 mM | 190 mM | 530 mM | 1.1 M | – | – | – | – |
| $B_{max2}$ | 0.5 | 0.5 | 0.5 | 0.5 | – | – | – | – |

| | Dual-topology channels | |
|---|---|---|
| | E88Q/SCN⁻ | E88Q/Cl⁻ |
| $K_{i,1}$ | 398 mM | 542 mM |
| $B_{max1}$ | 0.5 | 0.5 |
| $K_{i,2}$ | 1.2 M | 5.8 M |
| $B_{max2}$ | 0.5 | 0.5 |

**Table 3.** Fluc-Bpe inhibition and physical properties of halides and pseudohalides.

| | $K_{i,1}$ (mM) | $K_{i,2}$ (mM) | $K_i$ (oriented system, mM) | Radius (Å) | $pK_a$ | $\Delta G_{hyd}$ (kcal/mol) | $\Delta G_{Born}$ (kcal/mol) | Log $K^*_{Cl\text{-}X}$ |
|---|---|---|---|---|---|---|---|---|
| $F^-$ | – | – | – | 1.33 | 3.2 | −112 | −114 | −1.5 |
| $Cl^-$ | 137 | 1100 | 480 | 1.81 | −7 | −83 | −86 | 0 |
| $NO_3^-$ | 45 | 530 | – | 1.99 | −1.3 | −73 | −72 | 1.9 |
| $SCN^-$ | 9.0 | 190 | – | 2.49 | 1 | −69 | −63 | 3.23 |
| $OCN^-$ | 6.8 | 107 | 7.9 | 2.16 | 3.7 | −89 | −72 | 0.82 |

*Relative anion partition coefficient between water and PVC membrane, a measurement that reflects the lyotropic (Hofmeister) series, described in **Smith et al., 1999**.

on the S83C background and measured fluoride current inhibition by $Cl^-$ and $OCN^-$. The appreciable ~60-fold difference in $Cl^-$ and $OCN^-$ block characteristic of WT channels is almost eliminated for E88Q channels, which display only ~fourfold difference in $Cl^-$ and $OCN^-$ affinity (*Figure 4F*, *Figure 4—figure supplement 5*, *Table 2*). This effect is almost entirely due to the 10-fold less potent block of E88Q by $OCN^-$. Qualitatively similar results were obtained for $SCN^-$ block of randomly oriented WT and E88Q channels (*Figure 4—figure supplement 6*). Even if we are cautious in quantifying the effect because $Cl^-$ block cannot be measured to saturation, a qualitative reading of these experiments suggests that Glu88 contributes to anion recognition at the end of the pore defined by the T-E-Y triad.

## Discussion

### The vestibule end of the pore

In this work, we fused electrophysiology, X-ray crystallography, and liposome flux assays to identify the routes by which fluoride ions access the previously identified fluoride-binding sites along the polar track of Fluc homologs Fluc-Bpe and Fluc-Ec2. One anion-binding site, identified by the anomalous diffraction of $Br^-$ in the Fluc-Ec2 homolog, is located at the bottom of the electropositive vestibule and is sensitive to mutagenesis as well as modification of a nearby sidechain with the bulky thiol-reactive anion MTSES. Moreover, conversion of a serine from this anion-binding site to a cysteine introduces a strong pH-dependence to the fluoride channel activity, demonstrating that this position comprises part of the permeation pathway. Ion accumulation in aqueous entryways is a well-characterized feature of many ion channels, serving to increase the rate at which ions process to the constricted selectivity filter (*Doyle et al., 1998*; *Latorre and Miller, 1983*; *Payandeh et al., 2011*).

We speculate that the vestibule serine (S81 in Fluc-Ec2/S83 in Fluc-Bpe), which is absolutely invariant in Fluc channels, plays a central role in fluoride access to the dehydrated polar track. It is worth noting that a rotamerization of the vestibule serine would bring this sidechain within hydrogen-bonding distance of one such polar track fluoride position, F1 (*Figure 5*, right panel). A mechanism involving translocation of fluoride ions by rotamerization of amino acid sidechains lining the pore has been proposed for the Fluc channels previously and would be consistent with the measured conductance of these proteins (*Stockbridge et al., 2015*; *Last et al., 2017*). Since threonine enjoys less conformational flexibility than serine, such a mechanism might explain why S81T is non-functional in Fluc-Ec2 and why the Ser to Thr substitution has not arisen over evolutionary time in any Fluc channel. The hydrogen bond between the fluoride and the vestibule serine seems to be dispensable, and mutant channels with an alanine at the position retain robust fluoride currents. Similarly, conversion of polar track residues to alanine also had mild consequences for Fluc-Ec2 (*Last et al., 2017*). We note that, in experiments to monitor fluoride currents, especially single channels, saturating fluoride concentrations and high potentials are required due to the channels' relatively low conductance. We speculate that these mutants might have more drastic consequences at the low mM fluoride concentrations typical in the biological context.

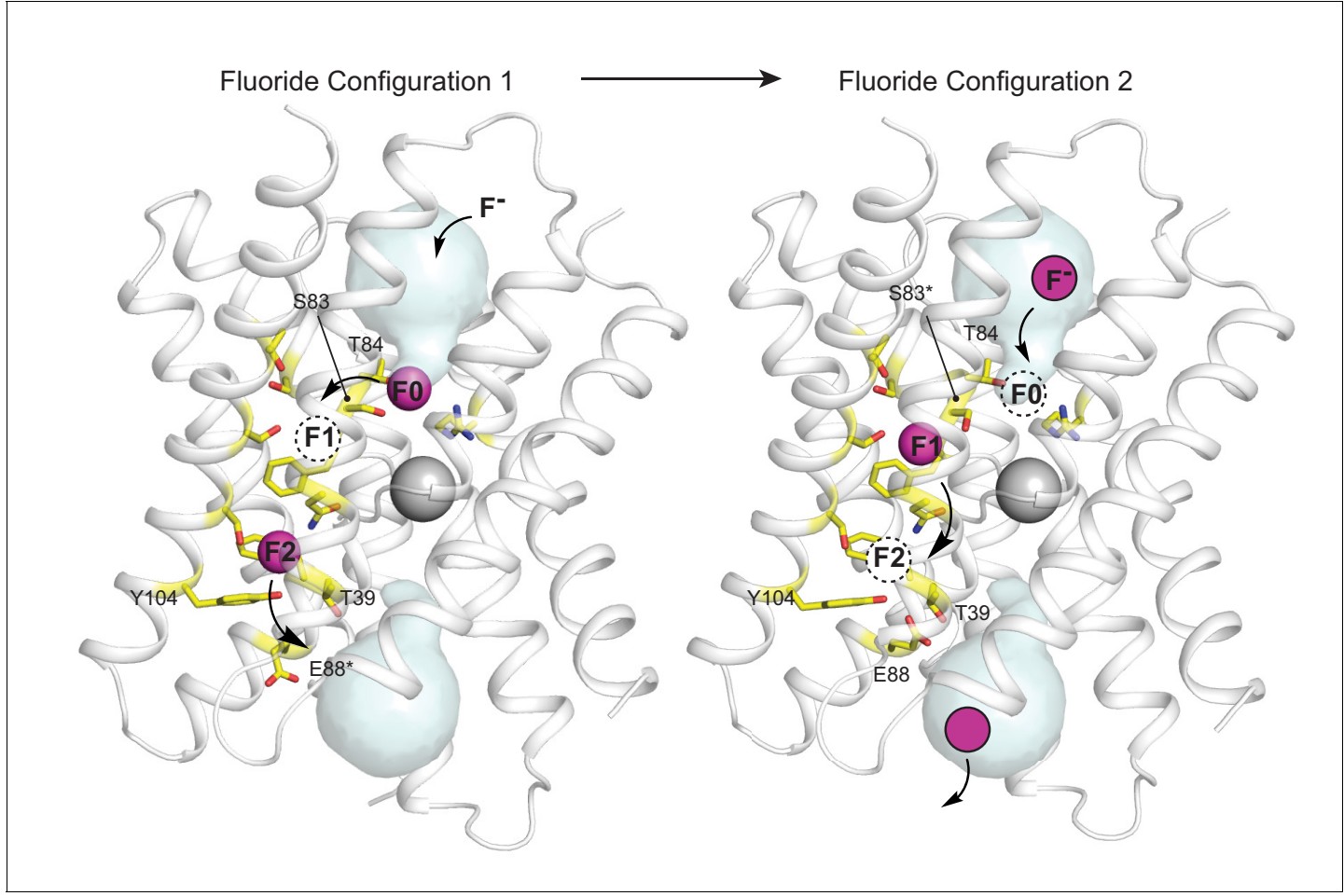

**Figure 5.** Proposed multi-ion permeation mechanism for Fluc-Bpe. For clarity, only one pore is shown. Cartoon structure is shown in transparent gray, aqueous vestibules are shown as pale cyan surfaces, and residues that have been shown to contribute to the pore (this work and references *Stockbridge et al., 2015*; *Last et al., 2016*; *McIlwain et al., 2020*; *Last et al., 2017*) are shown as yellow sticks. The five pore-lining residues identified in this work are labeled. Asterisks indicate that the rotamer shown is hypothetical and has not been observed crystallographically. Occupied fluoride ion sites are shown in pink, unoccupied fluoride-binding sites are shown as dashed circles, and the proposed movement of ions between binding sites is indicated by arrows.

The online version of this article includes the following figure supplement(s) for figure 5:

**Figure supplement 1.** Structure of Fluc-Bpe with all sidechains along the permeation pathway labelled.

## The T-E-Y triad end of the pore

Based on sequence analysis and site-directed mutagenesis, we have also identified the opposite end of the pore, which, in Fluc-Bpe, is defined by a hydrogen-bonded trio of conserved sidechains, T39, Y104, and E88, contributed by each of the three pore-lining helices. We propose that, in the resting state of the channel, the E88 carboxylate resides in the position observed in the crystal structures (*Figure 5*, right panel) (in structures, this position is additionally enforced by monobody binding). E88 is stabilized in this position by the positive dipole of helix 3b and hydrogen bond donors T39 and Y104, where it helps compensate the otherwise positive electrostatics of the unoccupied channel. We suggest that when F- is present, the permeant anion electrostatically repels the E88 carboxylate, perhaps competing for the same binding site at the top of helix 3b (*Figure 5*, left panel).

Other anions are also able to compete for this site in the permeation pathway, competitively inhibiting fluoride currents when bound. We observed that, for a series of halides and pseudohalides, the selectivity series is correlated to the $pK_a$ of the conjugate acid (*Table 3*, *Figure 4—figure supplement 2*); we propose that $pK_a$ is actually a proxy for the anion's strength as a hydrogen bond acceptor (basicity). Although $pK_a$ and basicity are not strictly correlated across anion types, the

properties are relatively well correlated within a single anionic series, such as the halide/pseudoha-lide series tested here (*Pike et al., 2017*; *Gilli et al., 2009*). Thus, we suggest that an anion's propensity to serve as a hydrogen bond acceptor contributes to its recognition by the Flucs, helping to explain the channel's remarkable indifference to Cl⁻, the fluoride ion's most biologically relevant competitor. In contrast to Cl⁻, and like OCN⁻, F⁻ is a famously strong hydrogen bond acceptor.

## Proposed mechanism of fluoride permeation

For all of Fluc's idiosyncrasies, we propose a mechanism with much in common with other well-characterized ion channels (*Figure 5*). The negative charge of the fluoride ions is counterbalanced by the protein's few positive charges, the vestibule arginines and the structural central Na⁺. Experiments have shown that both pores are functional for F⁻ permeation (*Last et al., 2016*), but it seems highly unlikely that all six anion positions (three anions in each of two pores) are simultaneously occupied. Rather, we imagine a scenario of alternating occupancy, as proposed for other multi-ion pores, in which a fluoride moving into one binding site electrostatically hastens its neighbor into the next position in the sequence. We propose either that the densities observed in the crystal structure represent partially occupied fluoride sites or that the monobodies used as crystallization chaperones alter the electrostatic landscape in the pore, increasing ion occupancy. Indeed, in crystal structures of Fluc-Bpe with a monobody occupying only one side of the channel, each pore contained only one fluoride density, rather than two, in the polar track (*McIlwain et al., 2018*).

In *Figure 5*, the starting configuration (left panel) shows an F⁻ bound in the site identified by anomalous scattering, at the bottom of the vestibule, labeled F0. We propose that as additional fluoride ions enter the electropositive vestibule, the fluoride ion at F0 is electrostatically repelled, providing energy for desolvation and translocation into the narrowest part of the channel at position F1 (*Figure 5*, right panel). But the F1 binding site is not pre-assembled: rotamerization of the vestibule serine (S83 in Fluc-Bpe), which is possible with serine but not threonine, accompanies the lateral movement of the anion. Other sidechains have also been proposed to adopt new rotameric conformations in order to ligand the anion at F1, including N43 (*Stockbridge et al., 2015*) and S84 (*Last et al., 2017*). Thus, we propose that the F1 binding site is assembled simultaneously with its occupation by fluoride. The rotamerization of channel sidechains to accompany ions through the pore has been proposed for other channels as well, including the Orai and voltage-gated calcium channels (*Hou et al., 2020*; *Sather and McCleskey, 2003*).

We imagine that this configuration is short-lived: a new fluoride ion settles into the deep vestibule F0 site, the fluoride at F1 moves farther down the channel to F2, and the S83 sidechain returns to its position facing the vestibule. The binding site at F2 is in close proximity to the E88 carboxylate; the electrostatic conflict could be resolved if E88 swings out into solution, allowing the fluoride at F2 to exit the channel, having now traversed the bilayer (whereby E88 could then resume its position at the pore exit without conflict). We have shown that the E88Q mutant reduces both fluoride currents and block of fluoride currents by OCN⁻. We propose that both behaviors arise because the mutant sidechain, which does not bear a negative charge, is not easily dislodged from the binding site via electrostatic conflicts with the permeant fluoride or the cyanate blocker.

This proposed mechanism introduces several previously unrecognized amino acids involved in fluoride permeation and extends the pathway to the aqueous solutions on both sides of the bilayer. It also explains the evolutionary conservation and physiological consequences of mutation described for conserved sidechains, including the invariant serine (position S81 in Ec2 or S83 in Bpe) and triad glutamate (position E88 in Bpe) (*Smith et al., 2015*; *Berbasova et al., 2017*). Also, while these experiments provide the first hints of a molecular mechanism for anion recognition by the Flucs, they also emphasize how robust the channel's anion selectivity is. Despite dozens of point mutations to two homologs, alone and in combination (summarized in *Figure 5—figure supplement 1* and *Table 4*), no mutant that permits the permeation of any other anion has been reported yet. It may be that there is no unique selectivity filter but that several regions of the channel work together to achieve selectivity, so that abolishing anion selectivity requires destruction of the channel itself. Alternatively, channel selectivity might be achieved by matching the number of available ligands in the pore to the preferred coordination number of the anion, as has been proposed for K⁺ channels (*Bostick and Brooks, 2007*; *Bostick and Brooks, 2009*). F⁻ is a superlative in this regard, requiring fewer ligands than any other anion. If this is the case, relaxing the selectivity might require adding coordinating ligands along the pore, which would be difficult to accomplish with site-

**Table 4.** Compiled results of anion transport experiments for Fluc-Bpe and Fluc-Ec2.
Results from Fluc-Ec2 are shown in italics, with numbering according to Fluc-Bpe for reference to the structure in *Figure 5—figure supplement 1*.

| Reference | Mutant (no F⁻ permeation) | Mutant (F⁻ permeation retained, no Cl⁻ permeation) |
|---|---|---|
| *Stockbridge et al., 2015* | | F82I, F85I, N43D |
| *Last et al., 2016* | F82I, F85I | |
| *Last et al., 2017* | F82Y, F82S, F82A, F82L, F82I, F82T, F85Y, F85S, F85A | S112A, T116V, T116I, S83A, F82M |
| *McIlwain et al., 2020* | | N43S, R22K |
| This work | S83T, S83C, T39A, T39V, T39C, T39N, E88K, Y104S, Y014H, Y104W, Y104I | S83A, S83A/S84A, Y104F, T39S, E88Q, E88D, E88A |

directed mutagenesis alone. Indeed, even accounting for the addition of coordinating ligands via sidechain rotamerization, the F1 and F2 sites have relatively small coordination numbers (~four including the phenylalanine ring edges). Chloride, in contrast, prefers at least six ligands in its coordination sphere (*Bostick and Brooks, 2009*; *Ohtaki and Radnai, 1993*; *Cametti and Rissanen, 2009*; *Merchant and Asthagiri, 2009*).

As a rare example of an anion channel required to select against the biologically dominant anion, the Fluc channels present an excellent case study of biochemical anion recognition. But the Fluc channel's stringent anion recognition, as quantified here, is physiologically essential, too. In electrophysiology experiments, in the presence of saturating 300 mM F⁻, the apparent $K_i$ values for block by Cl⁻ and other anions are correspondingly low. But in the bacterial cytoplasm, during an F⁻ challenge, with F⁻ ion between 100 μM and 10 mM (*Ji et al., 2014*), and Cl⁻ ion between 10 and 100 mM (*Schultz et al., 1962*), even a small increase in the inhibitory effects of Cl⁻ would represent a serious challenge to the efficacy of these channels and the survival of the bacteria.

# Materials and methods

**Key resources table**

| Reagent type (species) or resource | Designation | Source or reference | Identifiers | Additional information |
|---|---|---|---|---|
| Gene (*Bordetella pertussis*) | Fluc-Bpe | NCBI | WP_003818609.1 | Bears mutation R28K to increase yield (PMID:26344196) |
| Gene (*Escherichia coli* virulence plasmid) | Fluc-Ec2 | NCBI | WP_001318207.1 | Bears mutation R25K to increase yield (PMID:26344196). For cysteine modification experiments, C74A (this paper — see *Figure 1—figure supplement 1*). |
| Recombinant DNA reagent | Fluc-Bpe in pET21a (plasmid) | PMID:26344196 | | Expression vector for Fluc-Bpe. Available upon request. |
| Recombinant DNA reagent | Fluc-Ec2 in pET21a (plasmid) | PMID:26344196 | | Expression vector for Fluc-Ec2. Available upon request. |
| Chemical compound, drug | Isethionic acid | Wako Chemicals, Richmond VA | 107-36-8 | |
| Chemical compound, drug | MTSES | Toronto Research Chemicals | S672000 | |
| Chemical compound, drug | *E. coli* polar lipids | Avanti, Alabaster, AL | #100600C | |
| Chemical compound, drug | n-decyl-β -D-maltopyranoside | Anatrace, Maumee, OH | D322 | |

*Continued on next page*

*Continued*

| Reagent type (species) or resource | Designation | Source or reference | Identifiers | Additional information |
|---|---|---|---|---|
| Other | Monobodies S9 and S12 | PMID:25290819 | | Purified from *E. coli* according to the protocol described in the reference. PMID:25290819 |

## Chemicals and reagents

Potassium isethionate was prepared from isethionic acid (Wako Chemicals, Richmond, VA). Detergents were from Anatrace and lipids from Avanti Polar Lipids. MTSES ((2-sulfonatoethyl)methanethiosulfonate) was from Toronto Research Chemicals.

## Protein expression, purification, and reconstitution

Mutant channels were constructed using standard molecular biology techniques and verified by sequencing. All constructs bore functionally neutral mutations, R25K (Fluc-Ec2) or R28K (Fluc-Bpe), which increase protein yield (*Stockbridge et al., 2015*). Constructs that introduced a cysteine (Ec2-I48C and Ec2-S81C) also bore the mutation C74A. WT Fluc-Bpe is cysteine-free. Histidine-tagged Fluc-Bpe and Fluc-Ec2 were expressed in *E. coli* and purified via cobalt affinity chromatography according to published protocols (*Stockbridge et al., 2014*; *Stockbridge et al., 2015*; *McIlwain et al., 2020*). The buffer for the final size-exclusion step was 100 mM NaBr, 10 mM 2-[4-(2-hydroxyethyl)piperazin-1-yl]ethanesulfonic acid (HEPES), pH 7, for crystallography applications, or 100 mM NaCl, 10 mM HEPES, pH 7, for functional reconstitution. For reconstitution, proteins were mixed with detergent-solubilized *E. coli* polar lipids (Avanti Polar Lipids; 10 mg/ml) at a ratio of 0.1 µg protein/mg lipid for single-channel bilayer electrophysiology, 0.2 µg protein/mg lipid for liposome flux experiments, or 5 µg protein/mg lipid for macroscopic bilayer experiments. The protein/detergent/lipid mixture was dialyzed for 36 hr (6 l buffer per 50 mg lipid over three buffer changes). Proteoliposomes were stored at −80°C until use, at which point the suspension was freeze-thawed three times and extruded 21 times through a 400 nm filter to form liposomes.

## X-ray crystallography

After purification, monobody S9 and Fluc-Ec2 were mixed in a 1:1 molar ratio as described in *Stockbridge et al., 2015*. For Ec2-S81C, the protein mixture was used to set up sitting drop vapor diffusion crystal trays with a 1:1 mixture of protein solution and mother liquor. Crystals were formed in either 0.1 M glycine, pH 8.7–9.2, 31–36% polyethylene glycol (PEG) 600 or 0.1 M ammonium sulfate, 0.1 M N-(2-acetamido)iminoacetate (ADA), pH 6–6.5, 31–36% PEG 600 over 3–7 days and were frozen in liquid nitrogen prior to data collection at 13.5 keV at the Life Sciences Collaborative Access Team beamline 21-ID-D at the Advanced Photon Source, Argonne National Laboratory. Phases were calculated by molecular replacement with Phaser (*McCoy et al., 2007*) using Fluc-Ec2 and the monobody S9 as search models (pdb:5A43), followed by refinement with Refmac (*Murshudov et al., 2011*) and Phenix (*Liebschner et al., 2019*) and model building in real space with Coot (*Emsley et al., 2010*).

## Planar lipid bilayer electrophysiology

Experiments were performed as described previously (*Stockbridge et al., 2013*). Electrophysiological recordings were acquired at a holding voltage of −200 mV, electronically filtered at 1 kHz during acquisition, and digitally filtered to 500 Hz for analysis. Solutions in the cis and trans chambers varied as described in the text. Typical solutions contained 300 mM NaF with 10 mM 3-morpholinopropane-1-sulfonic acid (MOPS), pH 7. For MTSES and anion block experiments, the sodium salt of each anion was prepared as a concentrated solution in 300 mM NaF and 10 mM MOPS, pH 7, and added to the cis chamber with thorough manual mixing. The final MTSES concentration was 1 mM. For experiments in which the pH was varied, recording buffers additionally contained 10 mM 2-(N-morpholino)ethanesulfonic acid (MES, for pH 5.5 experiments) or 10 mM glycine (for pH 9 experiments). A pre-determined aliquot of dilute isethionic acid or NaOH was added to adjust the pH in the cis chamber, and the final pH value was confirmed after each experiment. Because hydrofluoric acid has

a $pK_a$ of 3.4 and is extremely hazardous, we avoided lowering the pH of fluoride solutions below 5.5. Macroscopic bilayer recordings shown are representative of three to seven independent bilayer experiments, and single-channel experiments are from 9 to 17 independent channel fusions for each mutant. All constructs used for electrophysiology experiments were purified from at least two independent protein preparations, and no prep-to-prep variation was observed.

## Fluoride efflux from liposomes

Fluoride efflux from liposomes was monitored using a fluoride-selective electrode as described previously (*Brammer et al., 2014*). Intraliposomal solution contained 300 mM KF, 10 mM $Na^+$ isethionate, 10 mM HEPES-KOH, pH 7. The external solution was exchanged by passing liposomes over a Sephadex G-50 spin column equilibrated in 300 mM $K^+$ isethionate, 10 mM Na isethionate, 10 mM HEPES-KOH, pH 7. Proteoliposomes were diluted 20-fold in matching buffer and fluoride efflux initiated by addition of 1 μM valinomycin. At the end of the experiment, remaining encapsulated fluoride was released from the liposomes by addition of 50 mM n-octyl-β-D-glucoside. Fluoride efflux was normalized against total encapsulated fluoride. In most cases, the result of this assay was binary: either the mutant had no activity relative to background leak (<100 ions/s) or the rate of fluoride efflux exceeded the response time of the electrode (>$10^4$ ions/sec). Efflux experiments were performed three to six independent times, with replicates derived from at least two independent protein preparations. In all cases of a binary result (no activity or >$10^4$ ions/sec), all replicates were in

**Table 5.** Liposome efflux experiments: compiled results from all replicates.

| Construct | Anion | Figure | Rate (ions/s): Prep 1 | Rate (ions/s): Prep 2 | Mean ± SEM |
|---|---|---|---|---|---|
| Ec2 WT | $F^-$ | 1-S1 | >$10^4$, >$10^4$ | >$10^4$, >$10^4$ | >$10^4$ |
| Ec2 C74A | $F^-$ | 1-S1 | >$10^4$, >$10^4$ | >$10^4$, >$10^4$ | >$10^4$ |
| Ec2 WT | $F^-$ | 2a | >$10^4$, >$10^4$ | >$10^4$, >$10^4$ | >$10^4$ |
| Ec2 S81A | $F^-$ | 2a | >$10^4$, >$10^4$, >$10^4$ | >$10^4$, >$10^4$, >$10^4$ | >$10^4$ |
| Ec2 S81T | $F^-$ | 2a | <100, <100 | <100, <100 | <100 |
| Ec2 S81C | $F^-$ | 2a | <100, <100 | <100, <100 | <100 |
| Ec2 S81A/S82A | $F^-$ | 2a | 8860, 6400 | 9640, 7840, 8860 | 8320 ± 560 ions/sec |
| Ec2 S81A/S82A | $Cl^-$ | 2-S2 | <50, <50 | <50, <50 | <50 |
| Bpe S83A/T84A | $Cl^-$ | 2-S2 | <50, <50 | <50, <50 | <50 |
| Bpe T39V | $F^-$ | 3-S1 | <100, <100 | <100, <100 | <100 |
| Bpe T39S | $F^-$ | 3-S1 | >$10^4$, >$10^4$, >$10^4$ | >$10^4$, >$10^4$ | >$10^4$ |
| Bpe T39C | $F^-$ | 3-S1 | <100, <100, <100 | <100, <100 | <100 |
| Bpe T39A | $F^-$ | 3-S1 | <100, <100 | <100, <100 | <100 |
| Bpe T39N | $F^-$ | 3-S1 | <100 | <100, <100 | <100 |
| Bpe E88A | $F^-$ | 3-S1 | >$10^4$, >$10^4$ | >$10^4$, >$10^4$, >$10^4$ | >$10^4$ |
| Bpe E88Q | $F^-$ | 3-S1 | >$10^4$, >$10^4$ | >$10^4$, >$10^4$, >$10^4$ | >$10^4$ |
| Bpe E88D | $F^-$ | 3-S1 | >$10^4$, >$10^4$ | >$10^4$, >$10^4$, >$10^4$ | >$10^4$ |
| Bpe E88K | $F^-$ | 3-S1 | <100 | <100, <100 | <100 |
| Bpe Y104F | $F^-$ | 3-S1 | >$10^4$, >$10^4$, >$10^4$ | >$10^4$, >$10^4$ | >$10^4$ |
| Bpe Y104S | $F^-$ | 3-S1 | <100, <100, <100 | <100, <100 | <100 |
| Bpe Y104H | $F^-$ | 3-S1 | <100, <100 | <100, <100 | <100 |
| Bpe Y104I | $F^-$ | 3-S1 | <100, <100 | <100 | <100 |
| Bpe I50W | $F^-$ | 1-S1 | 600, 720, 960 | 650, 550, 720 | 700 ± 60 ions/sec |
| Bpe Y104F | $Cl^-$ | 4-S1 | <50, <50 | <50, <50 | <50 |
| Bpe T39S | $Cl^-$ | 4-S1 | <50, <50 | <50, <50 | <50 |
| Bpe E88Q | $Cl^-$ | 4-S1 | <50, <50 | <50, <50 | <50 |
| Bpe E88A | $Cl^-$ | 4-S1 | <50, <50 | <50, <50 | <50 |

agreement (*Table 5*). Light-scattering experiments (*Figure 4—figure supplement 3*) were performed as previously described (*Stockbridge et al., 2012*). Proteoliposomes containing 300 mM KF, KCl, or KOCN and 10 mM HEPES, pH 7, were diluted in assay buffer (300 mM $K^+$ isethionate, 10 HEPES, pH 7). 90° light scattering was monitored at 550 nm upon addition of valinomycin (0.1 µg/ml final concentration).

## Acknowledgements

We thank the LS-CAT beamline staff for technical assistance, Christian Macdonald for assistance with sequence analysis, and members of the Stockbridge lab for comments on the manuscript and project. We are grateful to José Faraldo-Gómez and Robyn Stix (NIH/NHLBI) for insightful conversations about channel electrostatics.

## Additional information

### Competing interests

Randy B Stockbridge: Reviewing editor, *eLife*. The other authors declare that no competing interests exist.

### Funding

| Funder | Grant reference number | Author |
| --- | --- | --- |
| National Institutes of Health | R35-GM128768 | Randy B Stockbridge |

The funders had no role in study design, data collection and interpretation, or the decision to submit the work for publication.

### Author contributions

Benjamin C McIlwain, Conceptualization, Formal analysis, Investigation, Visualization, Methodology, Writing - original draft; Roja Gundepudi, B Ben Koff, Investigation; Randy B Stockbridge, Conceptualization, Formal analysis, Supervision, Funding acquisition, Visualization, Methodology, Writing - original draft, Project administration, Writing - review and editing

### Author ORCIDs

Randy B Stockbridge (iD) https://orcid.org/0000-0001-8848-3032

### Decision letter and Author response

Decision letter https://doi.org/10.7554/eLife.69482.sa1
Author response https://doi.org/10.7554/eLife.69482.sa2

## Additional files

### Supplementary files

• Transparent reporting form

### Data availability

Atomic coordinates for the Fluc-Ec2 and mutants in the presence of Br- have been deposited in the Protein Data Bank under accession numbers 7KKR (WT); 7KKA (S81A); 7KKB (S81C); 7KK8 (S81T); 7KK9 (S81A/T81A). Source data files have been provided for all figures. No custom code was used.

The following datasets were generated:

| Author(s) | Year | Dataset title | Dataset URL | Database and Identifier |
| --- | --- | --- | --- | --- |
| McIlwain BC, | 2021 | Fluoride channel Fluc-Ec2 wild- | https://www.rcsb.org/ | RCSB Protein Data |

| | | | | |
|---|---|---|---|---|
| Stockbridge RB | | type with bromide | structure/7KKR | Bank, 7KKR |
| McIlwain BC, Stockbridge RB | 2021 | Fluoride channel Fluc-Ec2 mutant S81A with bromide | https://www.rcsb.org/structure/7KKA | RCSB Protein Data Bank, 7KKA |
| McIlwain BC, Stockbridge RB | 2021 | Fluoride channel Fluc-Ec2 mutant S81C with bromide | https://www.rcsb.org/structure/7KKB | RCSB Protein Data Bank, 7KKB |
| McIlwain BC, Stockbridge RB | 2021 | Fluoride channel Fluc-Ec2 mutant S81T with bromide | https://www.rcsb.org/structure/7KK8 | RCSB Protein Data Bank, 7KK8 |
| McIlwain BC, Stockbridge RB | 2021 | Fluoride channel Fluc-Ec2 mutant S81A/T82A with bromide | https://www.rcsb.org/structure/7KK9 | RCSB Protein Data Bank, 7KK9 |

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
