## [Decision Letter]

**Acceptance summary:**

This manuscript will be of interest to scientists in physiology and membrane biophysics. Using functional tests guided by new structures that reveal ion-binding sites, the authors propose an elegant permeation mechanism that helps explain the unusually high fluoride selectivity of the microbial "Fluc" ion channels.

**Decision letter after peer review:**

Thank you for submitting your article "The fluoride permeation pathway and anion recognition in Fluc family fluoride channels" for consideration by *eLife*. Your article has been reviewed by 3 peer reviewers, including Merritt Maduke as the Reviewing Editor and Reviewer #1, and the evaluation has been overseen by Kenton Swartz as the Senior Editor. The following individual involved in review of your submission has agreed to reveal their identity: Rachelle Gaudet (Reviewer #3).

Essential revisions:

1) The electrophysiology experiments with MTS reagents and the I48C mutation are lacking a control experiment with a positively charged MTS reagent such as MTSET. For the experiment to confirm that the MTS is inhibiting through the electropositive vestibule, it should be shown that a positively charged MTS reagent either doesn't inhibit or inhibits with significantly slower kinetics.

2) Related to the previous point, Figure 1d shows only "before and after" from a single

experiment with current before MTS addition, and a lower current after, with the actual reaction kinetics not shown. We suspect this part of the trace may have been removed due to stirring artifacts, though this is not stated. The interpretation of the result will be much more compelling if you could show the kinetics of the MTS reagent. This should be possible using a lower concentration of MTS reagent (in the μM range, prepared just before use). We note that Figure 2e shows a reasonable timecourse when changing pH.

3) The summary data for experiments of Figure 1d would be best shown in the Figure (perhaps in a bar graph) and not just in the legend.

4) In the Methods section, it is stated that all electrophysiology experiments were performed on at least two independent protein preparations. However, the data as presented do not reveal the prep-to-prep variability. We recommend that the summary data (Figure 1d, Supp Figure 1, Supp Figure 2, etc etc) include tabulated data that indicate the results specifically for different preparations.

5) For Supplementary Figure 2, we would prefer to see the actual data, and not just single summary points, as the current traces are wobbly and therefore it is not immediately clear how one picks a value for each pH condition. For Figure 2e, a bar graph or some other method of presenting the summary data should be included.

6) For the experiments in Figure 2a, summary data should be shown in addition to the representative data.

7) A minor shortcoming regards the effects of pH on the mutations of Glu88 (p6, bottom). It is noted that the difference between Glu and Gln at position 88 are minimal, yet the lack of pH sensitivity between pH 7 and 8.7 is interpreted as reflecting a deprotonated Glu88. We would suggest instead that the glutamate is likely protonated, leading to the lack of difference between Glu and Gln, but that it's pKa is sufficiently shifted that it remains nearly fully protonated through the tested pH range. Such shifts in pKa are not unusual, especially in membrane proteins.

8) Which of the changes in Figure 3 are statistically significant?

9) The methods mention that the "Ec2-S81C construct also bore a mutation C74A", but this is not mentioned elsewhere and not data are presented to evaluate the function of the C74A background. The authors should provide information about how mutation of this residue, which contacts the bound Br-, does or does not impact ion channel function.

10) The pH dependence of the S81C and S83C mutations are used to propose that the cysteine is likely to be readily deprotonated. It could strengthen your arguments to generate predicted pKa values based on their structures. (The same suggestion applies to the arguments about E88.) One way to do this is to use PROPKA.

11) In the validation reports for Ec2-S81T, one of the Br- ion (residue # 2) has a very high B-factor and poor fit to the map (RSCC and RSR). This prompted a closer look at the models and maps. Looking at the Ec2-S81T map, it seems like Br- ion 2 is not positioned at the peak of density, and would likely benefit from more careful refinement, especially because it is a crucial part of this model. There is also a very strong unmodeled spherical density (>5 σ in the 2Fo-Fc map and >10 σ in the Fo-Fc map for chain C) right next to S81 of each monobody. (Ec2-S81C and WTBr have similar unmodeled blobs.)

Figures:

1) The arrows and arrowheads in all figures should be carefully checked to make sure that they line up with the actual time of addition (valinomycin and detergent) and buffer switches.

2) Figure 2d: using purple for the Br- ion is very confusing with the F- ions almost the same color. They should be consistently represented in the same orange color as in the other figures in the manuscript.

3) Figure 3a: It is unclear how residues were chosen for yellow highlighting, as there are plenty of other residues in the figure that seem just as or more conserved as the highlighted residues. Also, it would be helpful if residue numbers for the Bpe sequence were included in the figure. Finally, the "blue" highlight is dark enough that the black letters are difficult to make out.

4) Figure 3d: can the region of the recordings corresponding to the insets be indicated in the corresponding top traces?

5) Supp Figure 12: A line pointing to F82 would be useful.

Reviewers Combined Review:

This manuscript describes experiments aimed at understanding the ion-permeation pathway in the unusual and interesting Fluc family of microbial fluoride-selective ion channels. Previously, the channel pathway had been postulated based on locations of fluoride ions seen in crystal structures, but the tightly packed nature of the protein precluded certainty on this point. Here, the authors present four new crystal structures of the *E. coli* Fluc channel in the presence of the anomalous scattering ion, bromide, providing provide strong evidence for previously uncharacterized ion binding site one of the channel entry vestibules.

The authors follow up this structural evidence with mutagenesis of nearby residues. They test the role of a residue adjacent to the density by mutating it to Cysteine and reacting with a negatively charged MTS reagent, which reduces ion conduction as measured electrophysiologically. Other mutations in that and nearby residues also reduce ion conduction. Particularly compelling results are that the low ion conduction can be overcome for the *E. coli* S81C (and B. pertussis S83C) variants by lowering the pH, suggesting that the channel permeation pathway is very sensitive to local changes at that site. To identify potential anion entryways at the other end of the channel, the authors perform a sequence alignment of relatively distant family members and find several candidate residues whose roles in conductance are analyzed using a set of elegant selectivity and block assays.

Overall, the work presented here is very nice and the interpretations are solidly based on the data. However, some aspects of the experiments need to be clarified and extended. The experiments with MTS reagents are missing a critical control; characterization of the C74A background mutation is missing; and some experimental details of reproducibility and are not fully reported.

[Editors' note: further revisions were suggested prior to acceptance, as described below.]

Thank you for resubmitting your work entitled "The fluoride permeation pathway and anion recognition in Fluc family fluoride channels" for further consideration by *eLife*. Your revised article has been reviewed by 2 peer reviewers and the evaluation has been overseen by Kenton Swartz as the Senior Editor, and a Reviewing Editor.

The Reviewers agree that you have substantially improved the manuscript, especially with respect to reproducibility and the presentation of the data in most cases. The Reviewers are not completely satisfied with your argument against the need for the MTSET control: since there are two other cysteines in the protein, the control should be performed. Nevertheless, given that WT-like construct is not affected by MTS, which provides some reasonable support for the idea that the introduced cysteine is indeed the target, the additional control is not required. Similarly, though the reviewers agree that the MTS experiment would have been better with lower concentrations (so that the reaction is not over by the end of the stirring), such experiments are not essential to your conclusions and therefore will not be required. The inclusion of the currents during stirring is helpful, as is the coloring scheme to deemphasize that part of the data. Since the records are now continuous, we request that you remove the "break" marks in the figure.

---

## [Author Response]

Essential revisions:1) The electrophysiology experiments with MTS reagents and the I48C mutation are lacking a control experiment with a positively charged MTS reagent such as MTSET. For the experiment to confirm that the MTS is inhibiting through the electropositive vestibule, it should be shown that a positively charged MTS reagent either doesn't inhibit or inhibits with significantly slower kinetics.

We do not agree with the reviewers that MTSET modification is an essential control. In this construct, there is a single modifiable cysteine, I48. When the cysteine is present, currents are blocked by MTSES addition. When the cysteine is removed, currents are completely insensitive to MTSES addition. We have a high-resolution structure that shows that residue 48 is located in the vestibule, and that the vestibule is electropositive. We can’t imagine any other interpretation of the data except that MTSES inhibits by modifying the cysteine introduced at position 48 in the vestibule. It’s a reasonable prediction that MTSET would have slower kinetics of modification, but we do not think this experiment would serve as a control for any plausible alternative possibility. In addition, the interpretation that fluoride ions enter the Fluc channels through the vestibule is bolstered by additional lines of evidence, including crystallography data, I50W mutant in the Bpe homologue, mutagenesis experiments, and S81C pH sensitivity experiments.

2) Related to the previous point, Figure 1d shows only "before and after" from a singleexperiment with current before MTS addition, and a lower current after, with the actual reaction kinetics not shown. We suspect this part of the trace may have been removed due to stirring artifacts, though this is not stated. The interpretation of the result will be much more compelling if you could show the kinetics of the MTS reagent. This should be possible using a lower concentration of MTS reagent (in the μM range, prepared just before use). We note that Figure 2e shows a reasonable timecourse when changing pH.

We have updated our figures throughout the manuscript so that the electrical noise during stirring is shown in a light gray color. The kinetic response after MTSES addition is visible through the noise.

3) The summary data for experiments of Figure 1d would be best shown in the Figure (perhaps in a bar graph) and not just in the legend.

We have added a bar graph to Figure 1d.

4) In the Methods section, it is stated that all electrophysiology experiments were performed on at least two independent protein preparations. However, the data as presented do not reveal the prep-to-prep variability. We recommend that the summary data (Figure 1d, Supp Figure 1, Supp Figure 2, etc etc) include tabulated data that indicate the results specifically for different preparations.

We have color-coded the points in the bar graphs reported in Figure 1d, 2e, and 3c to indicate which measurements come from which biochemical prep. For liposome efflux experiments of point mutants, we thought that the most efficient way to convey this information was in a Table summarizing each result. We have added a new table (Table 5) that summarizes all replicates, including information about biochemical prep, for Figures 2a, 2b, 1-S1, 1-S2, 2-S2, 3-S1, 4-S1. For the experiments reported in Figure 4, we have included a supplementary figure (4-S3) that shows titrations of both the E88Q and WT constructs with Cl^-^ and OCN- with datapoints color-coded by biochemical prep. For the traces shown in 2-S1, the prep is indicated on the graph. Source data is also color coded according to biochemical prep.

None of the constructs reported in this manuscript show prep-to-prep variability.

5) For Supplementary Figure 2, we would prefer to see the actual data, and not just single summary points, as the current traces are wobbly and therefore it is not immediately clear how one picks a value for each pH condition. For Figure 2e, a bar graph or some other method of presenting the summary data should be included.

We have included full traces replicates of this experiment in 2-S1 and added a panel with single summary points to Figure 2e.

6) For the experiments in Figure 2a, summary data should be shown in addition to the representative data.

For most of the mutants, the fluoride transport rate is either above or below the limit of detection (>10^4^ ions per second and <100 ions/second, respectively). This is described in the Methods. We have updated this text slightly to include the lower limit of measurement:

“Fluoride efflux was normalized against total encapsulated fluoride. In most cases, the result of this assay is binary: either the mutant has no activity relative to background leak (<100 ions/s) or the rate of fluoride efflux exceeds the response time of the electrode (>10^4^ ions/sec). In all cases of a binary result (no activity or >10^4^ ions/sec) all replicates were in agreement.”

Of 27 different constructs tested in fluoride or chloride efflux experiments, 25 are either above or below the limit of detection. For these constructs, there is no way to calculate a mean or standard error for replicates – in other words, no way to summarize. (Ec2-S81A/S82A and Bpe-I50W are exceptions that fall within the dynamic range of the electrode.) The representative traces shown in Figure 2a (and 1-S1, 1-S2, 2-S2, 3-S1, 4-S1) are sufficient to convey this qualitative point (does a particular mutant have F- channel activity, yes or no?). We have added table in the supplement (Table 5) that shows the individual results from the replicates for each mutant. These are denoted “>10^4^” for measurements that exceed the response time of the electrode, “<100” for measurements below the limit of detection, or, for S81A and I50W, individual measurements of the rate, and are broken down by protein prep.

7) A minor shortcoming regards the effects of pH on the mutations of Glu88 (p6, bottom). It is noted that the difference between Glu and Gln at position 88 are minimal, yet the lack of pH sensitivity between pH 7 and 8.7 is interpreted as reflecting a deprotonated Glu88. We would suggest instead that the glutamate is likely protonated, leading to the lack of difference between Glu and Gln, but that it's pKa is sufficiently shifted that it remains nearly fully protonated through the tested pH range. Such shifts in pKa are not unusual, especially in membrane proteins.

We agree that shifted pKa values are not unusual. However, a glutamate with a pKa shifted by almost 6 units to >9 would be fairly unusual. We performed the propKa analysis as suggested in point #10, which predicts a pKa for this glutamate of 5.7.

We would also like to point out that the single channel currents for E88 and E88Q *are* quite different (Figure 3C) – a difference we argue is both statistically and mechanistically significant. What doesn’t differ between E88 and E88Q is the modulation of those currents by pH. We interpret this to mean that E88 (like E88Q) is not protonatable between pH 7 and 9. Of the two possibilities (a pKa above 9, or a pKa below 7), the pKa below 7 is the most likely. The propka analysis nicely backs up that inference.

We have updated the text around E88 to read:

“Since changing the protonation state of an acidic sidechain along the permeation pathway would be expected to have substantial ramifications for fluoride currents, these experiments suggest that the protonation state of Glu88 does not change as the pH is increased from 7 to 8.7, and therefore that the pK_a_ of E88 falls below ~6.5 or above ~9. […] In agreement with this interpretation, Propka calculates an approximate pK_a_ for Glu88 of 5.7 [12].”

8) Which of the changes in Figure 3 are statistically significant?

Although we have recorded enough single channels that the 10-15% differences in conductance between WT, Y104F, T39S, and E88D are statistically significant, we do not wish to make the argument that these differences are mechanistically important.

In the text, we do argue that the difference between WT and E88Q is *mechanistically* significant, and, accordingly, the difference in single channel conductance is statistically significant with p<0.0001.

We have updated the text as follows:

“T39S, E88D, Y104F retained F^-^ conductance at least 75% of WT levels, and we do not interpret these differences as mechanistically important. In contrast, E88Q exhibited currents one fifth of the wildtype levels, a more substantial difference that is also statistically significant at p<.0001 (unpaired t-test).”

9) The methods mention that the "Ec2-S81C construct also bore a mutation C74A", but this is not mentioned elsewhere and not data are presented to evaluate the function of the C74A background. The authors should provide information about how mutation of this residue, which contacts the bound Br-, does or does not impact ion channel function.

Thank you for pointing out our oversight in describing this construct. We use the C74A background as a “minimal cys” construct for all Ec2 cysteine labelling experiments. (The construct is not truly cysless because C16 cannot be altered without destabilizing the protein. However, C16 is buried at the dimer interface and does not react with thiol reagents.) In F- efflux experiments, it behaves like WT. We have added data to show this (Figure 1-S1 and Table 5). This same minimal-cys construct was used as the background for both the I48C MTSES addition experiments and the S81C experiments.

We have added language to the describe this construct when we introduce the I48C MTSES addition experiments, where we use it as the background for the “WT” control:

“We performed these experiments on a C74A background, which has fluoride transport properties similar to the WT protein. […] However, this residue is buried at the interface of helices 1 and 1’, and does not react with thiol reagents in the folded protein.”

We do not think that the structural evidence supports the suggestion that C74 contacts the Br-. Especially in our higher resolution structures, it is evident from the electron density that the C74 sidechain is pointed away from the Br- and packed against helix 2. As an example, electron density from the S81T structure is shown in Author response image 1. The closest atom of C74 is 4.4 Å from the Br-, and the SH group is 5.5 Å from the Br-. Even accounting for the possibility that C74 could adopt a different rotamer outside the crystal, none of the probable rotamers bring it within coordination distance of the Br-.

**Author response image 1. sa2fig1:** 

10) The pH dependence of the S81C and S83C mutations are used to propose that the cysteine is likely to be readily deprotonated. It could strengthen your arguments to generate predicted pKa values based on their structures. (The same suggestion applies to the arguments about E88.) One way to do this is to use PROPKA.

Thank you for this suggestion. We have updated the text around S81C to read:

“We posit that the electropositive environment of the vestibule perturbs the cysteine pK_a_ such that it is deprotonated at the pH of these experiments (pH 9 in the crystal structure, and pH 7.5 in the liposome flux experiments). The pK_a_ prediction software PropKa reinforces this possibility, calculating an approximate pK_a_ value of ~6 for S81C in the crystal structure of this mutant [12]. To test this idea explicitly…”

We have updated the text around E88 to read:

“Since changing the protonation state of an acidic sidechain along the permeation pathway would be expected to have substantial ramifications for fluoride currents, these experiments suggest that the protonation state of Glu88 does not change as the pH is increased from 7 to 8.7, and therefore that the pK_a_ of E88 falls below ~6.5 or above ~9. […] In agreement with this interpretation, Propka calculates an approximate pK_a_ for Glu88 of 5.7 [12].”

11) In the validation reports for Ec2-S81T, one of the Br- ion (residue # 2) has a very high B-factor and poor fit to the map (RSCC and RSR). This prompted a closer look at the models and maps. Looking at the Ec2-S81T map, it seems like Br- ion 2 is not positioned at the peak of density, and would likely benefit from more careful refinement, especially because it is a crucial part of this model. There is also a very strong unmodeled spherical density (>5 σ in the 2Fo-Fc map and >10 σ in the Fo-Fc map for chain C) right next to S81 of each monobody. (Ec2-S81C and WTBr have similar unmodeled blobs.)

We have re-refined Ec2-S81T and improved the fit to the density. A new PDB validation report is attached.

We are aware of the unmodelled blob near the monobody in the maps. We re-sequenced the monobody to ensure that the residue at position 81 is indeed a serine, and it is. The unmodelled blob is more electron dense than a water, and it is not a bromide. It may be a sulfate anion from the crystallization buffer. However, since this blob is associated with the monobody rather than the channel, is well removed from the bromide-binding site in the vestibule, and because it is difficult to assign it with certainty in these 2.5-3 Å-resolution structures, we have elected not to assign the density.

Figures:1) The arrows and arrowheads in all figures should be carefully checked to make sure that they line up with the actual time of addition (valinomycin and detergent) and buffer switches.2) Figure 2d: using purple for the Br- ion is very confusing with the F- ions almost the same color. They should be consistently represented in the same orange color as in the other figures in the manuscript.3) Figure 3a: It is unclear how residues were chosen for yellow highlighting, as there are plenty of other residues in the figure that seem just as or more conserved as the highlighted residues. Also, it would be helpful if residue numbers for the Bpe sequence were included in the figure. Finally, the "blue" highlight is dark enough that the black letters are difficult to make out.4) Figure 3d: can the region of the recordings corresponding to the insets be indicated in the corresponding top traces?5) Supp Figure 12: A line pointing to F82 would be useful.

We have made all of the requested changes to the figures. With regards to point #3, we have clarified the color-coding in the legend:

“Invariant pore-lining residues are shown in yellow. Pore-lining residues that are conserved in only one pore of the eukaryotic FEX channels are highlighted in blue.”

[Editors' note: further revisions were suggested prior to acceptance, as described below.]

The Reviewers agree that you have substantially improved the manuscript, especially with respect to reproducibility and the presentation of the data in most cases. The Reviewers are not completely satisfied with your argument against the need for the MTSET control: since there are two other cysteines in the protein, the control should be performed. Nevertheless, given that WT-like construct is not affected by MTS, which provides some reasonable support for the idea that the introduced cysteine is indeed the target, the additional control is not required.

We thank the editors for recognizing that the proposed experiment is not essential. Although we appreciate the suggestion from the reviewers, we would like to further elaborate on our rationale for not performing the MTSET modification experiment. We hope that this explanation is satisfactory.

For reasons described below, we think that (1) the control that we have already performed accomplishes the goal of distinguishing between the two potential cysteines and (2) the outcome of the MTSET experiment would be difficult to interpret and would be less likely to distinguish between the cysteines.

For the trace in the left panel of Figure 1d (labelled Ec2-I48C), the protein has cysteines at positions 16 (buried in the structure) and I48C. For the trace in the right panel (labelled Ec2-WT), the protein has cysteines at position 16 only (as described in the text, the C16 sidechains are buried and cannot be removed without destabilizing the protein).

The structure-guided introduction of a cysteine at position I48 provides a strong prediction that the MTS-mediated inhibition occurs through this site in the vestibule. However, we absolutely agree with the reviewers that, by itself, the electrophysiological experiment with the I48C construct (left panel) does not establish that MTSES reacts with the cysteine at position I48C. Modification of the other cysteine, C16, is an unlikely but valid possibility that must be controlled for. We can also imagine that MTSES interacts with the protein as a non-specific anionic blocker, blocking currents without forming any covalent linkage.

The experiment shown in the right panel of Figure 1d controls for both of these possibilities. If the current reduction in the left panel was due to modification of C16, then we would see a similar inhibitory effect in the right panel, since C16 is present in both constructs. We don’t see this. The same logic also rules out the possibility that MTSES blocks non-specifically. Any other interpretation would need to invoke the idea that the Ec2-WT and Ec2-I48C constructs adopt substantially different conformations. We think this possibility is quite unlikely, since the I48C’s biochemical and functional properties resemble WT, and extensive structural and electrophysiological experiments targeted at identifying conformational changes for the Flucs have uniformly suggested that these constitutively open channels do not undergo significant structural rearrangements (Turman and Stockbridge, 2017, *JGP* 149: 511-522; McIlwain, Newstead and Stockbridge, 2018, *Structure* 26: 635-639).

The reviewers propose that modification with MTSET would provide an additional control to distinguish between modification at C16 and I48C. Although this experiment might provide additional evidence beyond the structure that the modified cysteine is in an electropositive environment, we do not think that this experiment would be a superior control to confirm that MTS-mediated inhibition occurs at I48C. Both residues are located near the center of the protein, and the environment is likely to be electropositive for both (see Author response image 2). Thus, changing the electrostatic properties of the MTS reagent would be unlikely to distinguish between modification of C16 and I48C in a clearly interpretable way. Although we thank the reviewers for the suggestion, and we would be eager to perform any experiment that would rule out likely alternative explanations for the results in Figure 1, we do not think that the proposed MTSET modification experiment would do this.

Thank you for considering our work.

**Author response image 2. sa2fig2:** The structure shown here is of the WT channel. The Br- ions are shown as brick red spheres. Positions that are cysteines in the MTS experiments are colored magenta (I48C and C16). The features that contribute to the electropositive character of the channel are the central sodium ion (purple sphere) and R22 (cyan). These positive charges are in close proximity to both I48C and C16, and are likely to electrostatically influence both positions.

Similarly, though the reviewers agree that the MTS experiment would have been better with lower concentrations (so that the reaction is not over by the end of the stirring), such experiments are not essential to your conclusions and therefore will not be required. The inclusion of the currents during stirring is helpful, as is the coloring scheme to deemphasize that part of the data. Since the records are now continuous, we request that you remove the "break" marks in the figure.

We’ve removed the “break” marks in Figure 1 and 2. Thanks for catching this.